# Dysfunctional neuroplasticity in newly arrived Middle Eastern refugees in the U.S.: Association with environmental exposures and mental health symptoms

**Bengt B. Arnetz**[1]*, **Sukhesh Sudan**[1], **Judith E. Arnetz**[1], **Jolin B. Yamin**[2], **Mark A. Lumley**[2], **John S. Beck**[3], **Paul M. Stemmer**[4], **Paul Burghardt**[5], **Scott E. Counts**[1,3], **Hikmet Jamil**[1]

1 Department of Family medicine, College of Human Medicine, Michigan State University, Grand Rapids, Michigan, United States of America, 2 Department of Psychology, Wayne State University, Detroit, Michigan, United States of America, 3 Department of Translational Neuroscience, Grand Rapids, Michigan, United States of America, 4 Institute of Environmental Health Sciences, Wayne State University, Detroit, Michigan, United States of America, 5 Department of Nutrition and Food Science, Wayne State University, Detroit, Michigan, United States of America

* arnetzbe@msu.edu

**Data Availability Statement:** All relevant data are within the manuscript and its Supporting Information files.

## Abstract

### Background

Psychological war trauma among displaced refugees is an established risk factor for mental health disorders, especially post-traumatic stress disorder (PTSD). Persons with trauma-induced disorders have heightened neuroplastic restructuring of limbic brain circuits (e.g., amygdala and hippocampus), which are critical factors in the pathophysiology of PTSD. Civilians in war are exposed to both psychological trauma and environmental hazards, such as metals. Little is known about the possible mental health impact from such environmental exposures, alone or in combination with trauma. It is of special interest to determine whether war exposures contribute to dysfunctional neuroplasticity; that is, an adverse outcome from sustained stress contributing to mental health disorders. The current study examined Middle Eastern refugees in the United States to determine the relationships among pre-displacement trauma and environmental exposures, brain derived neurotrophic growth factor (BDNF) and nerve growth factor (NGF)—two neurotrophins reported to mediate neuroplasticity responses to stress-related exposures—and mental health.

### Methods

Middle Eastern refugees (n = 64; 33 men, 31 women) from Syria (n = 40) or Iraq (n = 24) were assessed 1 month after arrival to Michigan, US. Participants were interviewed in Arabic using a semi-structured survey to assess pre-displacement trauma and environmental exposure, PTSD, depression, anxiety, and self-rated mental health. Whole blood was collected, and concentrations of six heavy metals as well as BDNF and NGF levels were determined. Because these two neurotrophins have similar functions in neuroplasticity, we

**Funding:** BA - NIMH/NIH R01MH085793 (National Institute of Mental Health/National Institute of health; https://urldefense.com/v3/__https://www.nimh.nih.gov/index.shtml__;!!HXCxUKc!hZZI7yglrK-DsE2aI5M2reRjmTrnyp8_ntSV3vuk8K0t0OjJI43sPaK42a-HVRQ$), NIEHS/NIH P30ES020957 (National Institute of environmental Health Sciences/National Institute of Health; https://urldefense.com/v3/__https://www.niehs.nih.gov/index.cfm__;!!HXCxUKc!hZZI7yglrK-DsE2aI5M2reRjmTrnyp8_ntSV3vuk8K0t0OjJI43sPaK4kLxB0Q0$) SEC - NIA/NIH P01014449 (National Institute on Aging/National Institute of Health; https://urldefense.com/v3/__https://www.nia.nih.gov/__;!!HXCxUKc!hZZI7yglrK-DsE2aI5M2reRjmTrnyp8_ntSV3vuk8K0t0OjJI43sPaK4QMyfUsI$) The funders had no role in study design, data collection and analysis, decision to publish, or preparation of the manuscript.

**Competing interests:** The authors have declared that no competing interests exist.

combined them to create a neuroplasticity index. Linear regression tested whether psychosocial trauma, environmental exposures and biomarkers were associated with mental health symptoms.

## Findings

The neuroplasticity index was associated with PTSD (standardized beta, $\beta = 0.25$, $p < 0.05$), depression (0.26, < 0.05) and anxiety (0.32, < 0.01) after controlling for pre-displacement trauma exposures. In addition, pre-displacement environmental exposure was associated with PTSD (0.28, < 0.05) and anxiety (0.32, < 0.05). Syrian refugees and female gender were associated with higher scores on depression (0.25, < 0.05; 0.30, < 0.05) and anxiety scales (0.35, < 0.01; 0.27, < 0.05), and worse on self-rated mental health (0.32, < 0.05; 0.34, < 0.05). In bivariate analysis, the neuroplasticity index was related to blood lead levels ($r = 0.40$; $p < 0.01$).

## Conclusions

The current study confirms the adverse effects of war trauma on mental health. Higher levels of biomarkers of neuroplasticity correlated with worse mental health and higher blood lead levels. Higher neurotrophin levels in refugees might indicate dysfunctional neuroplasticity with increased consolidation of adverse war memories in the limbic system. Such a process may contribute to psychiatric symptoms. Further research is needed to clarify the pathobiological mechanisms linking war trauma and environmental exposures to adverse mental health.

## Introduction

The Middle Eastern conflicts in Iraq and Syria have resulted in extensive human suffering and death, domestic and international displacement of civilians, and destruction of critical infrastructures and entire villages [1, 2]. War-associated trauma is a major risk factor for the development of mental health disorders, including post-traumatic stress disorder (PTSD), depression, and anxiety [3–7]. Prospective studies of newly arrived refugees suggest that both pre- and post-displacement stressors contribute to adverse health trajectories [8–11]. Little is known, however, about the possible implications of environmental exposures for mental health and the role of biomarkers reflective of disease mechanisms in refugee populations.

Prior studies have reported an association between environmental war-related exposures and adverse somatic health. For example, in a cross-sectional study of civilians and military personnel engaged in the first Gulf War (GW)–Operation Desert Shield/Desert Storm, 1990–1991, the closer a person had resided to Kuwait–the epicenter of the war- the higher the prevalence of self-reported environmental war exposure—burning open pits, smoke, and mustard gas [12]. Higher self-reported aggregate exposure frequency to such environmental factors was associated with a higher prevalence of chronic somatic disorders. To our knowledge, however, prior studies of refugees exposed to hazardous environments have not included measurement of participants' concentrations of known toxicants, such as heavy metals, and related them to neurobiological markers or mental health symptoms.

Trauma exposure is necessary but not sufficient for the development of stress-related mental health disorders, suggesting marked variation among individuals in trauma vulnerability

[13, 14]. Established determinants of such vulnerability include female gender, intellectual challenges, prior trauma (e.g., adverse childhood experiences), and peri-trauma psychophysiological reactions [15–18]. There is an increasing focus on the role of genetic, epigenetic, and inflammatory systems in the development of mental health disorders [19–23]. Recent studies implicate dysregulation in brain neuroplasticity in trauma-related disorders [21, 24, 25]; that is, the limbic system (e.g., amygdala and hippocampus) undergoes neuroplastic changes that result in dysfunctional memories of traumatic events with adverse cognitive, emotional, and behavior outcomes. The two major neurotrophins involved in neuroplasticity–brain derived neurotrophic factor (BDNF) and nerve growth factor (NGF)—have been reported to be increased in blood of persons suffering from PTSD [26, 27].

In the current study of displaced refugees from Iraq and Syria, who had newly arrived in the U.S., we studied the associations between pre-displacement trauma, environmental war exposures, neuroplasticity and mental health outcomes. We hypothesized that pre-displacement stressors both contributed to worse post-displacement mental health (symptoms of PTSD, anxiety, and depression) and increased peripheral blood levels of neuroplasticity-related neurotrophins as potential biomarkers for the development and progression of these psychopathological symptoms.

## Methods

From February 2016 to August 2016, newly arrived refugees from Iraq and Syria were informed about the research by Samaritas, one of Michigan's largest refugee resettlement agencies. Refugees interested in learning more about the research left their contact information with a representative, who forwarded it to the research team, who then scheduled a follow-up meeting with each refugee. Assessments took place at one of the Arab Community Center for Economic and Social Services (ACCESS) medical clinics that provide health and social services. ACCESS is a nonprofit organization focusing on community services, cultural and social entrepreneurship, healthy lifestyle, education, and philanthropy. Potential participants were provided detailed written and oral information about the study, in Arabic, by the study team, and written consent was obtained.

### Survey questions

Participants were interviewed by a research assistant using a semi-structured survey in Arabic. The survey contained previously used and largely validated and published items or scales [8, 11]. Demographics included age, gender, and country displaced from (Iraq or Syria). Exposure and mental health symptoms were assessed as follows:

Trauma exposure was assessed using the Arabic version of the Harvard Trauma Questionnaire [28, 29]. The respondents used dichotomous responses (yes/no) to respond to questions such as whether they had "witnessed someone being physically harmed (beating, knifing, etc.)." Trauma exposure scores were calculated as a sum of all responses with higher scores indicating higher trauma exposure.

Environmental exposure was assessed using a 12-item revised version of a scale that had previously been developed in interviews with Iraqi refugees and validated in two separate studies [12, 30]. The respondents were asked, for how many days the exposure lasted; "<5" (coded = 1), "6–30" (2), "31+" (3). If they had not knowingly been exposed to a specific environmental factor, they were assigned 0. The magnitude of environmental exposure was calculated by summing the responses on the 12 items. Higher scores indicate higher exposures.

Post-traumatic stress disorder (PTSD) was assessed using the PTSD Checklist–Civilian version [31]. Participants responded to a series of questions about how much they had been

disturbed by specific experiences during the last month, using a 5-point scale ranging from "not at all" to "extremely". Responses were summed so that higher scores indicate more PTSD symptoms.

Depression was assessed using the 7- item depression subscale of the Hospital Anxiety and Depression Scale. A typical item was "I feel cheerful", with 4 response alternatives ranging from "not at all" to "most of the time." [32] Responses were summed so that higher scores indicate higher number of depressive symptoms.

Anxiety was assessed using the 21-item Beck Anxiety Inventory [33]. A typical question asked how much they had experienced being, e.g., "unable to relax" during the last month. There were 4 response alternatives, "not at all" to "severely, it bothered me a lot." Responses were summed so that higher scores indicate higher anxiety.

Self-rated mental health was assessed by using a 1-item 5-point Likert-type response scale, ranging from excellent to poor. Higher ratings indicate worse mental health.

## Biological measures

Blood samples were collected by a phlebotomist following the completion of the interview. A total of 10 mL of blood was collected in an Ethylenediaminetetraacetic acid (EDTA) treated 10 mL glass vial. Samples were placed on ice and immediately transported to the laboratory. Blood was then centrifuged at 4˚C, plasma separated into aliquots, and kept at -80˚C until later assessment.

## Neuroplasticity markers

Human-specific 96-well plate ELISAs were used to measure plasma concentrations of BDNF (OriGene #EA100205, Rockville, MD) or NGF (Aviva Systems Biology #OKEH00186, San Diego, CA). Duplicate samples were quantified by optical densitometry at 450 nm using a standard curve based on calibrators of known concentration. The intra-assay % CV for BDNF was 5.68% and the lower detection limit was 31.35 pg/mL. For NGF, the intra-assay % CV was 4.52% and the lower detection limit was 31.25 pg/mL. BDNF and NGF were summed to create the neuroplasticity index.

## Metal analyses

Whole blood concentrations of manganese (Mn), cobalt (Co), nickel (Ni), arsenic (As), cadmium (Cd) and lead (Pb) were measured at the Lumigen Instrument Center at Wayne State University. Isotope dilution mass spectrometry [34] was implemented to determine absolute quantitation of toxic metals and metalloids in the blood samples using the Agilent 7700x ICP-MS. Whole blood was diluted 1:1 with 0.1% TritonX-100 then further diluted with 8 volumes of 2% nitric acid for a 10-fold dilution. Proteins that precipitate upon addition of the nitric acid were removed by centrifugation. Final dilution of supernatants was made with 2% nitric acid to achieve a final 50-fold dilution. A six-point standard curve was generated using a mixed standard containing all the metals of interest diluted in 2% nitric acid. The standard curve was run before and after the analysis of the samples. Each sample was analyzed three times and the results averaged. Replicate %CV values for the six metals were: Mn, 3.5; Co, 29.5; Ni, 8.7; As, 7.4; Cd, 9.9; Pb, 30.8.

## Statistical analyses

Statistical analysis was conducted using IBM SPSS statistics, V.25, 2018 (IBM Corp, Armonk, NY). Scales and biological measures were tested for normality and z-transformed to allow for

aggregation of neuroplasticity measures as well as comparing beta values. Student's t-test and Chi square tests were used to test for significance between continuous and discrete variables, respectively. Bivariate analyses were used to determine correlations between self-reported exposures, biomarkers and mental health symptoms. The primary theoretical models assumed that pre-displacement exposures and neuroplasticity biomarker scores represented independent variables and the respective mental health symptoms, e.g., depression, as well as the mental health score, represented dependent variables. However, since this is a cross-sectional study, where the temporal relationship between variables cannot be established, we also tested reverse models using neuroplasticity scores as the dependent variable. The respective models' overall explained variances were used to determine the best performing models. Separate linear regression analyses were used to identify factors associated with PTSD, depression, anxiety, and self-rated mental health. In the first step, gender, age, and country being displaced from were entered; second step included pre-displacement trauma and environmental exposure; the final step included neuroplasticity score (the sum of z-scored BDNF and z-scored NGF). In addition, separate linear regression analyses assessing factors associated with neuroplasticity were run in 3-steps. The same variables mentioned above were entered in the first and second step. In the third and final step, the four mental health outcomes (PTSD, depression, anxiety and self-rated mental health) were entered separately. Significance was set at a two-sided p-value of $<0.05$ for all analyses.

The study protocol was approved by the Wayne State University Institutional Review Board.

## Results

The study involved 64 newly arrived Middle Eastern refugees from Iraq (24; 12 men and 12 women) and Syria (40; 21/19) with a mean age of 37.63 years (SD = 11.88; no significant differences between countries). Table 1 depicts mean and dispersion measures for self-reported pre-displacement trauma and environmental exposures, plasma levels of neuroplasticity biomarkers, blood levels of heavy metals, and scores on scales assessing mental health symptoms. Displaced refugees from Syria scored worse on depression, anxiety and self-rated mental health and had higher blood manganese levels as compared to Iraqi refugees.

Table 2 depicts bivariate correlations among trauma and environmental exposures, biological indices and mental health symptoms. Pre-displacement trauma was associated with PTSD scores, depression, anxiety and lower self-rated mental health. Pre-displacement environmental exposure was associated with pre-displacement trauma, PTSD, depression, and anxiety. The neuroplasticity index correlated with PTSD, depression, anxiety, self-rated mental health and blood lead levels.

### Regression modeling of mental health outcomes

In the full regression models shown in Table 3, pre-displacement trauma, pre-displacement environmental exposure and the neuroplasticity index were positively associated with symptoms of PTSD. Pre-displacement trauma, neuroplasticity, female gender and being from Syria rather than Iraq were positively associated with both depression and anxiety. However, self-reported pre-displacement environmental exposure was also positively associated with anxiety. The only significant factors associated with self-rated mental health were pre-displacement trauma, female gender and being from Syria. The explained variances for each model were 38% for PTSD, 47% for both depression and for anxiety, and 36% for self-rated mental health.

**Table 1. Descriptive characteristics of the study sample.**

| | All refugees (n = 64) | Iraqi (n = 24) | Syrian (n = 40) |
|---|---|---|---|
| Demographics | | | |
| Age (years); Mean (SD) | 37.63 (11.78) | 36.46 (12.16) | 38.33 (11.64) |
| Gender; n (%) | | | |
| Females | 31 (48.4) | 12 (50) | 19 (47.5) |
| Males | 33 (51.6) | 12 (50) | 21 (52.5) |
| Self-reported exposures; Mean (SD) | | | |
| Pre-displacement trauma (possible scale range 0 to 39); ($\alpha$ = .87) | 12.09 (6.02) | 10.74 (4.80) | 12.88 (6.54) |
| Pre-displacement environmental exposure (0 to 36); ($\alpha$ = .76) | 2.72 (4.39) | 3.58 (5.09) | 2.20 (3.88) |
| Neuroplasticity biomarkers (pg/ml); Mean (SD) | | | |
| Brain-derived neurotrophic factor | 1918.51 (1641.51) | 1541.29 (1394.41) | 2078.83 (1727.15) |
| Neurotrophic growth factor | 80.07 (72.39) | 70.89 (70.37) | 83.97 (73.77) |
| Neuroplasticity index | 1998.58 (1697.17) | 1612.18 (1459.64) | 2162.80 (1780.20) |
| Heavy metals concentration (ppb); Mean (SD) | | | |
| Lead | 0.64 (1.46) | 0.54 (1.03) | 0.69 (1.68) |
| Cobalt | 0.01 (0.00) | 0.01 (0.01) | 0.01 (0.00) |
| Manganese | 0.12 (0.08) | **0.08 (0.05)** | **0.14 (0.09)** [**] |
| Nickel | 0.11 (0.11) | 0.11 (0.02) | 0.12 (0.13) |
| Arsenic | 0.02 (0.01) | 0.02 (0.02) | 0.01 (0.01) |
| Cadmium | 0.02 (0.02) | 0.01 (0.02) | 0.02 (0.01) |
| Mental health symptoms; Mean (SD) | | | |
| Depression (0–21); ($\alpha$ = .86) | 6.31 (4.35) | **4.75 (3.79)** | **7.25 (4.44)** [*] |
| Anxiety (0–63); ($\alpha$ = .92) | 13.02 (11.30) | **8.79 (8.15)** | **15.55 (12.23)** [*] |
| Post-traumatic stress disorder (17–85); ($\alpha$ = .82) | 33.95 (12.41) | 30.62 (12.88) | 35.95 (11.83) |
| Mental health score; 1 (excellent) to 5 (poor) | 3.00 (1.11) | **2.38 (1.10)** | **3.38 (0.95)** [***] |

[*]p<0.05

[**]p<0.01

[***]p<0.001; boldface indicates significant differences b/w Iraqi and Syrian refugees; SD, standard deviation; ppb, parts per billion; pg/ml, picograms per milliliter; $\alpha$, Cronbach's alpha

## Regression modelling of neuroplasticity

In a series of reverse modeling, where neuroplasticity was used as the dependent factor, and PTSD, depression, anxiety, and self-rated mental health, respectively, as independent variables, there were significant associations as well (Table 4). However, the explained variances of each of the models were substantially smaller:18% with PTSD as an independent variable, 20% with depression, 24% with anxiety and 16% with self-rated mental health. Of note, neither demographics, pre-displacement trauma nor environmental exposure were significant in the final models, with the exception of age, which was significantly associated with the model that included self-rated mental health.

## Discussion

The current study focused on understanding the role of pre-displacement trauma and environmental exposures on the mental health of newly arrived Middle Eastern refugees in the United States. Furthermore, we were interested in exploring the possible roles of war-associated environmental exposures as well as biological pathways related to neuroplasticity in mental health. These biological processes have been implicated in stress-related mental health disorders in

**Table 2. Bivariate correlations b/w self-reported exposures, biomarkers and mental health symptoms (n = 64).**

| | 1 | 2 | 3 | 4 | 5 | 6 | 7 | 8 | 9 | 10 | 11 | 12 | 13 | 14 | 15 |
|---|---|---|---|---|---|---|---|---|---|---|---|---|---|---|---|
| 1 Trauma (Pre) | - | | | | | | | | | | | | | | |
| 2 Env. exposure | **0.40**** | - | | | | | | | | | | | | | |
| 3 BDNF | 0.08 | 0.04 | - | | | | | | | | | | | | |
| 4 NGF | 0.15 | -0.03 | **0.76***** | - | | | | | | | | | | | |
| 5 Neuroplasticity | 0.12 | 0.00 | **0.94***** | **0.94***** | - | | | | | | | | | | |
| 6 Lead | -0.04 | 0.10 | **0.36**** | **0.38**** | **0.40**** | - | | | | | | | | | |
| 7 Cobalt | -0.15 | 0.12 | -0.11 | -0.19 | -0.16 | **0.28*** | - | | | | | | | | |
| 8 Manganese | -0.15 | -0.17 | -0.19 | **-0.29*** | -0.26 | -0.16 | 0.10 | - | | | | | | | |
| 9 Nickel | -0.24 | -0.03 | 0.09 | 0.03 | 0.07 | -0.03 | 0.14 | -0.05 | - | | | | | | |
| 10 Arsenic | -0.16 | 0.16 | -0.04 | -0.18 | -0.12 | 0.24 | **0.71***** | 0.13 | 0.05 | - | | | | | |
| 11 Cadmium | 0.03 | 0.19 | 0.09 | 0.25 | 0.18 | 0.19 | **0.38**** | 0.14 | -0.03 | **0.32*** | - | | | | |
| 12 Depression | **0.50***** | **0.31**** | **0.28*** | **0.32*** | **0.32*** | 0.12 | 0.05 | -0.09 | -0.17 | 0.11 | 0.09 | - | | | |
| 13 Anxiety | **0.48***** | **0.28*** | **0.34**** | **0.37**** | **0.38**** | 0.13 | -0.08 | -0.11 | -0.15 | -0.16 | 0.07 | **0.63***** | - | | |
| 14 PTSD | **0.48***** | **0.34**** | **0.29*** | **0.29*** | **0.31*** | 0.08 | -0.03 | -0.03 | -0.17 | 0.05 | 0.04 | **0.64***** | **0.67***** | - | |
| 15 Mental health[1] | **0.32*** | 0.06 | **0.31*** | 0.25 | **0.30*** | 0.21 | 0.15 | 0.09 | 0.01 | 0.01 | 0.15 | **0.55***** | **0.47***** | **0.46***** | - |

*p<0.05

**p<0.01

***p<0.001; BDNF, Brain-derived neurotrophic factor; NGF, Nerve growth factor; Neuroplasticity, sum of z-scored BDNF and NGF; PTSD, post-traumatic stress disorder; [1]higher values indicate worse scores on self-rated mental health.

non-refugees and rodents [35, 36]. Finally, we were interested in assessing whether, controlling for known mental health risk factors, there were country-specific differences in exposures, metal burden, biomarkers and mental health outcomes. Any such differences might suggest unique and country-specific risk factors that might shed further light on refugee mental health. The studied refugees were newly arrived in the U.S. which eliminates the challenges in most post-displacement refugee mental health work to disentangle the effects of pre-displacement from post-displacement trauma and stressor exposures [10, 37].

This study is unique in terms of refugee health for three specific reasons: 1. We included self-rated as well as biological measures of environmental exposures. 2. We included biomarkers of neuroplasticity; 3. We studied consistency of findings across two different but simultaneously displaced groups of war-exposed persons.

Our primary models consistently revealed, as predicted, that pre-displacement trauma exposure was associated with post-displacement instrument based as well as self-rated mental health. Higher exposure to pre-displacement trauma was associated with more severe mental health symptoms but also with reporting higher exposure to war-associated environmental toxicants. In prior work, we did present an association between place of residence during the first Gulf War and self-reported war-associated environmental exposures as well as chronic somatic disorders [12].

Interestingly, when controlling for pre-displacement exposures, higher levels of BDNF and NGF were associated with worse scores on validated scales for PTSD, depression, and anxiety. As two of the brain's most critical neurotrophins, BDNF and NGF support a multitude of neuronal processes including neuroplasticity, which is the process by which the strength of synaptic connections between neurons is altered dynamically by cognitive challenges such as learning and memory [38]. Among learning and memory processes, emotional memories related to fear, stress, and trauma are critical to the development of disorders such as depression, anxiety, and PTSD [39]. Notably, BDNF and NGF mediate neuroplastic alterations

**Table 3. Linear regression for factors associated with PTSD, depression, anxiety and mental health ratings (n = 64).**

| | β | β | β |
|---|---|---|---|
| **Separate model for each of the mental health outcome** | **Step 1** | **Step 2** | **Step 3** |
| PTSD | | | |
| Age | 0.15 | 0.15 | 0.08 |
| Females (Ref. males) | -0.06 | 0.18 | 0.17 |
| Syrian refugees (Ref. Iraqi refugees) | 0.17 | 0.21 | 0.20 |
| Pre-displacement trauma | | **0.38**[**] | **0.35**[*] |
| Pre-displacement environmental exposure | | 0.28 | **0.28**[*] |
| Neuroplasticity | | | **0.25**[*] |
| $R^2$ | 0.06 | 0.32 | 0.38 |
| Depression | | | |
| Age | 0.10 | 0.10 | 0.03 |
| Females (Ref. males) | 0.01 | **0.26**[*] | **0.25**[*] |
| Syrian refugees (Ref. Iraqi refugees) | **0.29**[*] | **0.31**[*] | **0.30**[*] |
| Pre-displacement trauma | | **0.47**[***] | **0.44**[**] |
| Pre-displacement environmental exposure | | 0.24 | 0.24 |
| Neuroplasticity | | | **0.26**[*] |
| $R^2$ | 0.10 | 0.41 | 0.47 |
| Anxiety | | | |
| Age | 0.09 | 0.09 | 0.00 |
| Females (Ref. males) | 0.09 | **0.36**[**] | **0.35**[**] |
| Syrian refugees (Ref. Iraqi refugees) | 0.24 | **0.29**[*] | **0.27**[*] |
| Pre-displacement trauma | | **0.41**[**] | **0.38**[**] |
| Pre-displacement environmental exposure | | **0.32**[*] | **0.32**[*] |
| Neuroplasticity | | | **0.32**[**] |
| $R^2$ | 0.08 | 0.38 | 0.47 |
| Mental health ratings[1] | | | |
| Age | 0.07 | 0.07 | 0.01 |
| Females (Ref. males) | 0.19 | **0.33**[*] | **0.32**[*] |
| Syrian refugees (Ref. Iraqi refugees) | **0.38**[**] | **0.35**[**] | **0.34**[**] |
| Pre-displacement trauma | | **0.39**[**] | **0.36**[**] |
| Pre-displacement environmental exposure | | 0.02 | 0.02 |
| Neuroplasticity | | | 0.22 |
| $R^2$ | 0.18 | 0.32 | 0.36 |

[*]$p<0.05$

[**]$p<0.01$

[***]$p<0.001$; PTSD, post-traumatic stress disorder; β, standardized beta; [1]higher values indicate worse scores on self-rated mental health.

underlying these memory subtypes. For instance, mice heterozygous for BDNF display deficits in the acquisition of fear [40], whereas mice overexpressing BDNF receptors display enhanced fear [41], in contextual fear conditioning behavioral paradigms. Likewise, aggression and psychosocial stress behavioral tests both result in increased brain NGF expression [42, 43]. In addition, anxiogenic stimuli have been shown to differentially induce BDNF and NGF in the hippocampus and amygdala, two key structures associated with the consolidation of emotional memory [44]. Taken together, these data suggest that neurotrophin-mediated neuroplasticity in brain regions of critical importance for incorporating traumatic and stressful memories,

**Table 4. Linear regression for factors associated with neuroplasticity (n = 64).**

| | β | β | β |
|---|---|---|---|
| Separate models with each mental health symptom as independent variable | Step 1 | Step 2 | Step 3 |
| Age | **0.30**[*] | **0.30**[*] | 0.25 |
| Females (Ref. males) | 0.00 | 0.04 | -0.02 |
| Syrian refugees (Ref. Iraqi refugees) | 0.05 | 0.04 | -0.03 |
| Pre-displacement trauma | | 0.12 | -0.01 |
| Pre-displacement environmental exposure | | -0.01 | -0.10 |
| PTSD | | | **0.33**[*] |
| $R^2$ | 0.09 | 0.11 | 0.18 |
| Age | **0.30**[*] | **0.30**[*] | 0.25 |
| Females (Ref. males) | 0.00 | 0.04 | -0.06 |
| Syrian refugees (Ref. Iraqi refugees) | 0.05 | 0.04 | -0.08 |
| Pre-displacement trauma | | 0.12 | -0.07 |
| Pre-displacement environmental exposure | | -0.01 | -0.10 |
| Depression | | | **0.39**[*] |
| $R^2$ | 0.09 | 0.11 | 0.20 |
| Age | **0.30**[*] | **0.30**[*] | 0.25 |
| Females (Ref. males) | 0.00 | 0.04 | -0.13 |
| Syrian refugees (Ref. Iraqi refugees) | 0.05 | 0.04 | -0.09 |
| Pre-displacement trauma | | 0.12 | -0.07 |
| Pre-displacement environmental exposure | | -0.01 | -0.15 |
| Anxiety | | | **0.46**[**] |
| $R^2$ | 0.09 | 0.11 | 0.24 |
| Age | **0.30**[*] | **0.30**[*] | **0.27**[*] |
| Females (Ref. males) | 0.00 | 0.04 | -0.05 |
| Syrian refugees (Ref. Iraqi refugees) | 0.05 | 0.04 | -0.06 |
| Pre-displacement trauma | | 0.12 | 0.01 |
| Pre-displacement environmental exposure | | -0.01 | -0.01 |
| Mental health ratings [1] | | | 0.28 |
| $R^2$ | 0.09 | 0.11 | 0.16 |

[*]$p < 0.05$

[**]$p < 0.01$; PTSD, post-traumatic stress disorder; β, standardized beta; [1]higher values indicate worse self-rated mental health.

e.g., hippocampus, amygdala and pre-frontal cortex, contribute to PTSD and associated stress-related disorders (e,g., depression, anxiety), and that circulating levels of BDNF and NGF may reflect these brain alterations. In this regard, a recent study demonstrated that plasma BDNF levels are increased in US military officers with PTSD compared to unaffected officers and that BDNF Val66Met polymorphism modulated PTSD severity [26]. Hence, our systematic findings suggest that the observed upregulation of neuroplasticity is related to consolidating adverse experiences and memories resulting in PTSD, as well as depression and anxiety in the studied refugees [45]. We speculate that the positive association between neuroplasticity and worse mental health reflects a dysfunctional neuroplastic process. Moreover, there is a complex, context-dependent interaction between mature neurotrophins or their pre-processed proneurotrophin moieties and their cognate receptors, which can promote neuronal maintenance/survival or cell death, respectively [36, 46]. Whereas our ELISA assays could not differentiate between pro- and mature BDNF or NGF levels in plasma, further research will be

required to elucidate potential shifts in the stoichiometry of these neurotrophin moieties and the mechanistic contributions of differential neurotrophin signaling to adverse events such as PTSD.

Of further interest in support of the dysfunctional neuroplasticity hypothesis is the finding that concentrations of blood lead levels correlate with neuroplasticity scores.

We suggest that in addition to current best practice of assessing pre-displacement trauma exposure and post-displacement mental health in refugees, considerations should be given to assessment of self-rated and biological measures of environmental exposures of concern, as well as determining circulating plasma levels of neurotrophins. It would be relevant to determine the predictive value of such an expanded assessment panel in predicting refugees at high risk of developing post-displacement trauma disorders in their host country.

In the linear regression modeling of depression, anxiety and self-rated mental health, country remained a significant predictor after taking other risk factors into account. In cross-country comparisons, refugees from Syria scored higher on the depression and anxiety scales. They also had significantly higher levels of manganese as compared to persons displaced from Iraq. Therefore, we suggest further, and more definite studies in terms of the possible role of environmental exposures and its interaction with trauma exposures in the development of post-displacement mental health disorders in refugees. This is especially important when considering the high likelihood that displaced refugees end up living under socioeconomically strained and environmentally unhealthy conditions [9, 47].

The current study adds to the complexity of understanding mechanisms involved in the pathophysiology of trauma-associated mental health symptoms. Specifically, it suggests there is a need to better capture the total environmental exposure of displaced refugees, not merely psychological trauma. Second, by measuring biomarkers of neurotrophins and relating those to mental health scores, it might be possible to focus early interventions on refugees identified as being at heightened risk for the development of post-trauma disorders.

There are several limitations to this study. The study design is cross-sectional and causal relationships cannot be determined, although the models using mental health factors as dependent variables performed better than models using neuroplasticity as the dependent variable. The study was limited to only 64 displaced persons. We asked the participants to report trauma and environmental exposures of the past. We do not know whether recollection of events and exposures are accurate, neither do we have information on how far back such exposures might have occurred. We collected biomarkers at only one time. This limits the ability to delineate whether changes in biomarkers are temporary or more sustained. Additionally, it may not be possible to assess the neurotrophins and other biomarkers in refugees in low- and middle-income countries due to limited financial resources. However, if findings from this study are confirmed in future studies, the benefits of assessing these biomarkers might outweigh the financial burden since in the long run, it may save the costs associated with treating these mental health disorders.

In conclusion, the current study suggests that, in addition to assessing trauma exposures, there is a need to better define environmental exposures in persons with a history of having been exposed to war and conflicts. There might have been hitherto unknown environmental exposures that contributed to overall worse mental health outcomes in displaced Syrian refugees as compared to Iraqis. It is suggested that there is a dysfunctional neuroplastic process in trauma-exposed refugees with implications for post-displacement mental health and well-being.

## Supporting information

**S1 Dataset.**
(XLSX)

## Acknowledgments

Sincere thanks to our long-term community partners–Samaritas and ACCESS and to displaced persons from Iraq and Syria who, despite their tragedies and fight to reestablish their fractured lives in their new country, volunteered to take part in this study.

## Author Contributions

**Conceptualization:** Bengt B. Arnetz.

**Data curation:** Bengt B. Arnetz, Sukhesh Sudan, Jolin B. Yamin, Mark A. Lumley, Paul M. Stemmer, Hikmet Jamil.

**Formal analysis:** Bengt B. Arnetz, Sukhesh Sudan, Jolin B. Yamin, Mark A. Lumley, John S. Beck.

**Funding acquisition:** Bengt B. Arnetz.

**Investigation:** Jolin B. Yamin, John S. Beck, Paul M. Stemmer, Paul Burghardt, Scott E. Counts, Hikmet Jamil.

**Methodology:** Bengt B. Arnetz, Sukhesh Sudan, Judith E. Arnetz, Paul M. Stemmer, Scott E. Counts.

**Project administration:** Bengt B. Arnetz, Jolin B. Yamin, Mark A. Lumley.

**Resources:** Bengt B. Arnetz, Paul M. Stemmer, Paul Burghardt, Scott E. Counts.

**Software:** Bengt B. Arnetz, Sukhesh Sudan.

**Supervision:** Bengt B. Arnetz, Judith E. Arnetz, Mark A. Lumley, Scott E. Counts, Hikmet Jamil.

**Validation:** Bengt B. Arnetz, Jolin B. Yamin, Paul M. Stemmer.

**Visualization:** Bengt B. Arnetz, Sukhesh Sudan.

**Writing – original draft:** Bengt B. Arnetz, Sukhesh Sudan, Paul M. Stemmer.

**Writing – review & editing:** Sukhesh Sudan, Judith E. Arnetz, Jolin B. Yamin, Mark A. Lumley, John S. Beck, Paul M. Stemmer, Paul Burghardt, Scott E. Counts.

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
