## [Decision Letter · Decision Letter 0]

7 Feb 2020

PONE-D-19-29424

Dysfunctional neuroplasticity in newly arrived Middle Eastern refugees in the U.S.: Association with environmental exposures and mental health disorders

PLOS ONE

Dear Dr. Arnetz,

Thank you for submitting your manuscript to PLOS ONE. After careful consideration, we feel that it has merit but does not fully meet PLOS ONE’s publication criteria as it currently stands. Therefore, we invite you to submit a revised version of the manuscript that addresses the points raised during the review process.

The reviewers addressed some minor comments about your manuscript. Please revise your manuscript carefully.

We would appreciate receiving your revised manuscript by Mar 23 2020 11:59PM. To enhance the reproducibility of your results, we recommend that if applicable you deposit your laboratory protocols in protocols.io, where a protocol can be assigned its own identifier (DOI) such that it can be cited independently in the future. For instructions see: http://journals.plos.org/plosone/s/submission-guidelines#loc-laboratory-protocols

We look forward to receiving your revised manuscript.

Kind regards,

Kenji Hashimoto, PhD

Academic Editor

PLOS ONE

Reviewers' comments:

Reviewer's Responses to Questions

**Comments to the Author**

1. Is the manuscript technically sound, and do the data support the conclusions?

Reviewer #1: Yes

Reviewer #2: Yes

2. Has the statistical analysis been performed appropriately and rigorously? 

Reviewer #1: Yes

Reviewer #2: Yes

3. Have the authors made all data underlying the findings in their manuscript fully available?

Reviewer #1: Yes

Reviewer #2: Yes

4. Is the manuscript presented in an intelligible fashion and written in standard English?

Reviewer #1: Yes

Reviewer #2: Yes

5. Review Comments to the Author

Reviewer #1: The manuscript focussed an important issue, and it was well written. It will contribute much for mental health fields. The methodological and statistical is appropriate. The interpretation of the results is rational. I think it should be accepted in the present form.

Reviewer #2: This is an interesting paper where the authors have studied 64 newly arrived refugees from Syria and Iran living in Michigan USA. They were studied one month after their arrival by making an assessment on PTSD, depression, anxiety, self-reported mental health. neurotrophins- BDNF and NGF as proxy measures of neuroplasticity and exposure to heavy metals. The authors found independent associations between environmental exposures and neuroplasticity with different mental health problems. Few of the findings can be interpreted as different from what one expects.

As the authors argue this is a pioneer study where they have included two important measures, namely exposure to environmental toxins and neurobiological markers for measuring neuroplasticity in addition to classical descriptive studies in refugee trauma. This has certainly given an extra strength to this paper which will ultimately help us to understand refugee trauma mechanism better.

Abstract is well written and well presented. However, I wonder if the authors could avoid using the terminologies like predictor or predicted ( e.g. line 49 ) as this is not a longitudinal study and the direction of causality is thus not very obvious. I would suggest the authors to be a little bit careful when they speculate for their findings. In line 64-66, they argue for role of neutrophins as facilitator of apoptosis which is far away from existing knowledge and understanding of its roles.

Introduction. Well written with relevant literatures

Methods: Sample selection is somewhat limited and the sample size is also rather small. Choice of the instruments and statistical methods are quite well explained and good.

Results are presented well both in tables and texts. I am just curious to know if the authors have sometimes thought about using neuroplasticity as a dependent variable in their analyses. It could have been interesting to look that too.

Discussions are also well balanced and presented. However, I think that authors are quite optimistic for using sophisticated biological parameters is studying refugee trauma which is not always possible specially in low-and middle- income countries.

6. PLOS authors have the option to publish the peer review history of their article (what does this mean?). If published, this will include your full peer review and any attached files.

Reviewer #1: No

Reviewer #2: Yes: Suraj Bahadur Thapa

---

## [Author Response · Author response to Decision Letter 0]

19 Feb 2020

Response:

Thank you for alerting us to this requirement. We have now reviewed the template on the journal website and the manuscript has been formatted to meet the journals’ style requirements.

Response:

We will be uploading the de-identified dataset as supporting information files.

Reviewer #1: The manuscript focused an important issue, and it was well written. It will contribute much for mental health fields. The methodological and statistical is appropriate. The interpretation of the results is rational. I think it should be accepted in the present form.

Response:

We would like to thank the reviewer for reviewing the manuscript.

Reviewer #2: This is an interesting paper where the authors have studied 64 newly arrived refugees from Syria and Iran living in Michigan USA. They were studied one month after their arrival by making an assessment on PTSD, depression, anxiety, self-reported mental health. neurotrophins- BDNF and NGF as proxy measures of neuroplasticity and exposure to heavy metals. The authors found independent associations between environmental exposures and neuroplasticity with different mental health problems. Few of the findings can be interpreted as different from what one expects.

As the authors argue this is a pioneer study where they have included two important measures, namely exposure to environmental toxins and neurobiological markers for measuring neuroplasticity in addition to classical descriptive studies in refugee trauma. This has certainly given an extra strength to this paper which will ultimately help us to understand refugee trauma mechanism better.

Abstract is well written and well presented. However, I wonder if the authors could avoid using the terminologies like predictor or predicted (e.g. line 49 ) as this is not a longitudinal study and the direction of causality is thus not very obvious. I would suggest the authors to be a little bit careful when they speculate for their findings. In line 64-66, they argue for role of neutrophins as facilitator of apoptosis which is far away from existing knowledge and understanding of its roles.

Introduction. Well written with relevant literatures

Methods: Sample selection is somewhat limited and the sample size is also rather small. Choice of the instruments and statistical methods are quite well explained and good.

Results are presented well both in tables and texts. I am just curious to know if the authors have sometimes thought about using neuroplasticity as a dependent variable in their analyses. It could have been interesting to look that too.

Discussions are also well balanced and presented. However, I think that authors are quite optimistic for using sophisticated biological parameters is studying refugee trauma which is not always possible specially in low-and middle- income countries.

Response:

We would like to thank the reviewer for these valuable suggestions. We agree with the reviewer and have changed the terminology to make it clear that this is a cross-sectional study that cannot prove causality and therefore we have removed terms such as predictive in the Abstract and elsewhere. 

As per reviewer’s suggestion, we have now deleted the sentence in line 64-66 of the original Abstract that mentioned role of neutrophins as facilitator of apoptosis.

Since we had hypothesized neuroplasticity to be potential biomarkers for the development of mental health disorders (PTSD, depression and anxiety), we had used it as an independent variable. However, the reviewer raised an interesting point regarding using neuroplasticity as a dependent variable. We ran the requested analysis, revised the text and added a Table 4. Interestingly, in separate linear regressions, the three mental health disorders were each associated with neuroplasticity. However, the models explained substantially lesser variance compared to when they were run as per our proposed model. We thank the reviewer for suggesting this alternative causative model.

 We agree with the reviewer that the analysis of these biomarkers may not be possible in low-and middle-income countries due to the financial limitations. We have now acknowledged this as a potential limitation. However, we also believe that if our findings are confirmed in future studies, the costs for these analyses might be justified since measuring these biomarkers allows refugee mental health professionals to focus sparse resources on those with the largest needs.

---

## [Editor Report · Decision Letter 1]

20 Feb 2020

Dysfunctional neuroplasticity in newly arrived Middle Eastern refugees in the U.S.: Association with environmental exposures and mental health symptoms

PONE-D-19-29424R1

Dear Dr. Arnetz,

We are pleased to inform you that your manuscript has been judged scientifically suitable for publication and will be formally accepted for publication once it complies with all outstanding technical requirements.

With kind regards,

Kenji Hashimoto, PhD

Section Editor

PLOS ONE
---

## [Editor Report · Acceptance letter]

24 Feb 2020

PONE-D-19-29424R1 

Dysfunctional neuroplasticity in newly arrived Middle Eastern refugees in the U.S.: Association with environmental exposures and mental health symptoms 

Dear Dr. Arnetz:

I am pleased to inform you that your manuscript has been deemed suitable for publication in PLOS ONE. Congratulations! Your manuscript is now with our production department. 

With kind regards,

on behalf of

Prof. Kenji Hashimoto 

Section Editor

PLOS ONE